# Two Methods for Superposing the Structures of Like-Molecule Assemblies: Application to Peptide and Protein Oligomers and Aggregates

**DOI:** 10.3390/molecules30051156

**Published:** 2025-03-04

**Authors:** Adam Liwo, Mateusz Leśniewski

**Affiliations:** Faculty of Chemistry, University of Gdańsk, Fahrenheit Union of Universities, Wita Stwosza 63, 80-308 Gdańsk, Poland; m.lesniewski.714@studms.ug.edu.pl

**Keywords:** structure superposition, like-molecule assemblies, peptides and proteins, singular value decomposition, quaternions

## Abstract

Two algorithms are proposed for the superposition of assemblies of like molecules (e.g., peptide and proteins homooligomers and homoaggregates), which do not require examining all permutations of the molecules. Both start from searching the mutual orientation of the two assemblies over a grid of quaternion components for the sub-optimal mapping and orientation of the molecules of the second to those of the first assembly. The first one, termed Like-Molecule Assembly Distance Alignment (LMADA), uses Singular Value Decomposition to superpose the two assemblies, given the sub-optimal mapping. The second one, termed Like-Molecule Assembly Gaussian Distance Alignment (LMAGDA), minimizes the negative of the logarithm of the sum of the Gaussian terms in the distances between the corresponding atoms/sites of all pairs of molecules of the two assemblies in quaternion components, starting from those estimated in the first stage. Both algorithms yield as good or nearly as good superposition, in terms of root mean square deviation (RMSD), as examining all permutations to find the lowest RMSD. LMADA results in lower RMSDs, while LMAGDA in a better alignment of the geometrically matching sections of the assemblies. The costs of the proposed algorithms scale only with $N^{2}$, *N* being the number of molecules in the assembly, as opposed to N! when examining all permutations.

## 1. Introduction

Measures of the similarity of the structures of a given molecule or a molecular assembly are necessary to dissect the ensembles obtained in molecular simulations or generated by bioinformatics approaches (e.g., MassiveFold [1]) into families of similar structures and to compare the representative structures with their reference (usually experimental) counterparts. These measures are widely implemented in the Community-Wide Experiments on the Critical Assessment of Techniques for Protein Structure Prediction (CASP) [2]. Superposition of two corresponding structures to minimize their distance is the basis of most of these methods and is usually accomplished by using the Singular Value Decomposition (SVD) algorithm [3] on the inter-molecule distance matrix [4,5]. The commonly used software for molecule superposition are the Local–Global Alignment (LGA) [6,7] and Template Modeling Score (TM-score) [8]. Quaternion-based algorithms, as an alternative to those based on SVD, have also been developed [9,10].

Superposition of a pair of objects requires one-to-one mapping of the points (atoms or extended atoms for molecules or their assemblies) of one to those of the second of them. This mapping is trivial for single molecules or assemblies of all-unlike molecules (e.g., protein heterodimers). Conversely, for those of like molecules, all mapping of the molecules of one of the assemblies on those of the other one must, in principle, be examined to obtain the best superposition. For example, when two trimers of extended polypeptide chains forming a three-stranded β-sheet are superposed, mapping chains A, B, and C of the first trimer on chains A, B, and C of the second one, respectively, will not result in the best superposition if the sequence of the chains in the ladder is ABC in the first and BAC in the second trimer. All six permutations of chain IDs of the second trimer should thus be examined for the best superposition. For assemblies of *N* like molecules, the number of possible mappings (permutations) is N!, making the brute-force search for the best mapping of the molecules of one assembly onto those of the other one infeasible for larger *N*. With N=8, the execution time already becomes prohibitively long, even with parallel processing, when superposition is an element of clustering an ensemble of more than 1000 structures of a peptide or protein assembly.

The existing algorithms for the alignment of protein oligomers, e.g., MM-align [11] and US-align [12], focus on assemblies of large monomers, in which the concern is finding the matching sections of the chains. These algorithms use dynamic programming and heuristic search for matching chains and chain segments of the compared structures and subsequently minimize the TM-score. Related algorithms for large DNA and RNA structure alignment were also developed [13,14,15], in which the matching motifs are searched in the two compared structures. For the comparison of oligomer structures, measures of monomer interface similarity, namely the Interface Patch Similarity (IPS) and the Interface Contact Similarity (ICS), were introduced [16]. These measures do not require mapping of the monomers of the compared structures. These measures are used in the assessment of oligomer models in CASP exercises [16].

In this paper, we propose two superposition algorithms, which do not require examining all permutations, termed Like-Molecule Assembly Distance Alignment (LMADA) and Like-Molecule Assembly Gaussian Distance Alignment (LMAGDA), respectively. Both are based on scanning the orientation space of the assemblies in the first stage to estimate a reasonable mapping of the molecules of one on those of the second assembly and the initial orientation of the assemblies. In LMADA, superposition is then executed by means of SVD, using the molecule-to-molecule mapping found in the first stage. In LMAGDA, a target function, which is the negative of the logarithm of the sum of Gaussians in the distances of the points of the molecules of the first to the corresponding points of those of the second assembly, is minimized to find the optimal superposition. We demonstrate that both methods give comparable root mean square deviations (RMSDs) as the brute-force method based on examining all permutations. The use of LMADA results in lower RMSDs on average, while LMAGDA superposes more closely the similar parts of the two assemblies.

## 2. Results

In what follows, the RMSD corresponding to the best mapping of the molecules of the superposed on those of the reference structure is termed the reference RMSD, $RMSD_{r}$. This RMSD is obtained by exploring all permutations of the molecules of the latter with respect to those of the former. We term this superposition algorithm the reference algorithm (see Section 3.1). The RMSD resulting from using the reference algorithm is the lowest RMSD for a given pair of structures. The RMSD obtained from LMADA is referred to as $RMSD_{1}$, while $RMSD_{d}$ [Equation (Equation 11) of Section 3.1] and $RMSD_{\Phi }$ [Equation (Equation 13) of Section 3.1] are used as measures of the fitness of the structures in assessing LMAGDA. $RMSD_{r}$ is considered as a reference to compare with $RMSD_{1}$ and $RMSD_{d}$ in a quantitative way. Because $RMSD_{\Phi }$ has the meaning of the measure of the distance between well-overlapping sections of the structures, it is compared with $RMSD_{r}$ only qualitatively. Additionally, we consider the RMSD corresponding to simple superposition with matching chains of the superposed assembly to the chains with the same IDs of the reference assembly; this RMSD is denoted $RMSD_{s}$. The corresponding algorithm will be referred to as the simple algorithm.

The quantitative measures of the comparison of $RMSD_{1}$, $RMSD_{d}$, and $RMSD_{s}$ with $RMSD_{r}$ are the average differences, ${\bar{\Delta RMSD}}_{1}$, ${\bar{\Delta RMSD}}_{d}$, and ${\bar{\Delta RMSD}}_{s}$, and the average relative differences, $\bar{\left(\frac{\Delta RMSD_{1}}{RMSD_{r}}\right)}$, $\bar{\left(\frac{\Delta RMSD_{d}}{RMSD_{r}}\right)}$, and $\bar{\left(\frac{\Delta RMSD_{s}}{RMSD_{r}}\right)}$, of the respective RMSDs and the reference RMSD. It should be noted that these numbers are always non-negative. Likewise, we define the standard deviations of the differences and relative differences $\sigma_{\Delta RMSD_{1}}$, $\sigma_{\Delta RMSD_{d}}$, $\sigma_{\Delta RMSD_{s}}$, $\sigma_{\frac{\Delta RMSD_{1}}{RMSD_{r}}}$, $\sigma_{\frac{\Delta RMSD_{d}}{RMSD_{r}}}$, and $\sigma_{\frac{\Delta RMSD_{s}}{RMSD_{r}}}$. These quantities are defined by Equations (Equation 1)–(Equation 4).(1)$$\begin{array}{c} {\bar{\Delta RMSD}}_{x}=\frac{1}{N_{p}}\sum_{i=1}^{N_{p}}\left(RMSD_{xi}-RMSD_{ri}\right) \end{array}$$(2)$$\begin{array}{c} \bar{\left(\frac{\Delta RMSD_{x}}{RMSD_{r}}\right)}=\frac{1}{N_{p}}\sum_{i=1}^{N_{p}}\frac{RMSD_{xi}-RMSD_{ri}}{RMSD_{ri}} \end{array}$$(3)$$\begin{array}{c} \sigma_{\Delta RMSD_{x}}=\sqrt{\frac{1}{N_{p}-1}\sum_{i=1}^{N_{p}}{\left[\left(RMSD_{xi}-RMSD_{ri}\right)-{\bar{\Delta RMSD}}_{x}\right]}^{2}} \end{array}$$(4)$$\begin{array}{c} \sigma_{\frac{\Delta RMSD_{x}}{RMSD_{r}}}=\sqrt{\frac{1}{N_{p}-1}\sum_{i=1}^{N_{p}}{\left[\frac{RMSD_{xi}-RMSD_{ri}}{RMSD_{ri}}-\bar{\left(\frac{\Delta RMSD_{x}}{RMSD_{r}}\right)}\right]}^{2}} \end{array}$$
where $N_{p}$ is the number of superposed pairs and the subscript *x* denotes the kind of RMSD (*r*, *d*, or *s*).

We also calculated the percentages of similar structures rejected corresponding to the three superposition algorithms, $\%R_{s}$, $\%R_{1}$, and $\%R_{d}$. These quantities are defined by Equation (Equation 5).(5)$$\%R_{x}=\frac{N_{p}\left(RMSD_{x}\geq 6\;\overset{˚}{A}\;\mathrm{and}\;RMSD_{r}<6\;\overset{˚}{A}\right)}{N_{p}\left(RMSD_{r}<6\;\overset{˚}{A}\right)}\times 100\;\%$$
where $N_{p}\left(RMSD_{x}\geq 6\;\overset{˚}{A}\;\mathrm{and}\;RMSD_{r}<6\;\overset{˚}{A}\right)$ is the number of structures for which $RMSD_{x}$ ($RMSD_{1}$ for LMADA or $RMSD_{d}$ for LMAGDA) is equal to 6 Å or higher but their $RMSD_{r}$ is lower than 6 Å and $N_{p}\left(RMSD_{r}<6\;\overset{˚}{A}\right)$ is that of the pairs of structures for which the reference RMSD is lower than 6 Å. The 6 Å cut-off was adapted from the paper by Reva et al. [17], who demonstrated that two random structures are unlikely to superpose within 6 Å RMSD.

### 2.1. Importance of Correct Mapping of the Molecules of the Superposed
on Those of the Reference Ensemble

To realize the importance of the search for the molecule-to-molecule mapping giving the best or at least a sensible superposition, let us consider two structures from the UNRES-simulated tetramer of CysZ8 shown in Figure 1. In panel A of the Figure, the structures superposed by using the reference algorithm (which gives the lowest RMSD) are shown. In panel B of the Figure, the superposed structures are shown with chains colored according to sequence to show that the directions of the superposed chains match. In panel C, the structures are superposed with the simple algorithm (assuming that the chains of the reference and of the superposed structures are matched after their IDs). As shown, the simple algorithm results in a poor superposition.

In Figure 2A, a plot of the $RMSD_{s}$ (corresponding to mapping the chains with the same ID for both structures on each other) vs. $RMSD_{r}$ (the reference RMSD corresponding to the chain mapping resulting in the best superposition) is shown for CysZ8 tetramers, hexamers, and octamers. In Figure 2B, the $RMSD_{s}$ is plotted vs. $RMSD_{r}$ for the MassiveFold models of the sven CASP16 oligomeric targets. As can be seen, most of the values of $RMSD_{s}$ are usually significantly larger than those of $RMSD_{r}$.

For quantitative comparison, the measures of the deviation defined by Equations (Equation 1)–(Equation 4) are collected in Table 1. As shown in the Table, for the CysZ8 oligomers, the RMSDs are much higher than the reference RMSDs on average, and over 85% of pairs superposing within the 6 Å RMSD cut-off would be assessed as dissimilar. For the MassiveFold models, about 50% or more of the pairs would be falsely assessed as dissimilar with the simple superposition approach, except for T2234 and T2240 (trimers), for which only about 11% of such pairs are found. Specifically, for both hexamers (T2235 and T2270) the percentages of pairs wrongly assessed as dissimilar by the simple algorithm are about 90% or higher. It can, therefore, be concluded that superposing the two assemblies (no matter whether generated by simulation or by a bioinformatics approach) with the assumption that chains with the same ID can be matched leads to an unacceptably high rejection rate of the pairs of similar structures.

### 2.2. Tests of LMADA

The RMSDs calculated by using LMADA ($N^{2}$ effort) and those calculated by the reference algorithm (N! effort) for the tetramer, hexamer, and octamer pairs of CysZ8 obtained in UNRES/MREMD simulations and for the MassiveFold models are compared in Figure 3, while the measures of the differences of the reference RMSDs and those from LMADA are collected in Table 1. It can be seen from Figure 2 and Figure 3 and from Table 1 that the RMSDs from LMADA diverge much less from those obtained by exhaustive chain-mapping enumeration compared to those obtained by the simple algorithm. The percentages of pairs wrongly assessed as dissimilar drops to no more than 6.5% for the CysZ8 octamer (Table 1). It can therefore be concluded that LMADA can be applied with confidence to superposing like-molecule assemblies, for which the enumeration of all possible mappings of the molecules of the superposed assembly on those of the reference assembly is impractical or even impossible to apply due to a large number of permutations.

The timings of LMADA are compared to those of the reference (enumeration of all mappings) algorithm in Table 2. As can be seen, the reference algorithm is faster up to tetramers, for which the number of chain permutations is only 24. Consequently, executing SVD for every mapping of the chains of the superposed on those of the reference structure is less expensive compared to computing the distances between all chains of the superposed and those of the reference structure for all 374 orientations defined by the quaternion component grid (cf. Section 3.1). For the CysZ8 hexamers, LMADA is of comparable efficiency, and for T2235 and T2270, LMADA is faster compared to the reference algorithm. For the CysZ8 octamer, it is about 30 times faster. For the CysZ8 decamer, there is a 3-order-of-magnitude difference between the execution times of the two algorithms, and the reference algorithm becomes clearly too expensive to apply.

### 2.3. Tests of LMAGDA

The plot of $RMSD_{\Phi }$ resulting from the minimization of Φ of Equation (Equation 12) vs. $RMSD_{r}$ for the CysZ8 oligomers and that for the MassiveFold models of the seven CASP16 targets are shown in Figure 4A and Figure 4B, respectively.

It can be seen that $RMSD_{\Phi }$ does not differentiate many pairs with $RMSD_{r}<1$ Å. This is reasonable because a moderately large $\sigma =\sqrt{8}$ was selected in computing Φ [Equation (Equation 12)]. For $RMSD_{r}$ between 1 and 6 Å, $RMSD_{\Phi }$ increases with increasing $RMSD_{r}$, but for $RMSD_{r}$ around 7 Å  pairs with low $RMSD_{\Phi }$ appear, especially for the CysZ8 oligomers (Figure 4A). In Figure 5, one such pair of structures of the CysZ8 tetramer is shown after superposition with LMADA (panels A and B) and LMAGDA (panels C and D). It can be seen from the figure that, although the two structures form a β-sheet ladder composed of extended chains, the two middle chains of the superposed structure run in an opposite direction to that of the chains of the reference structure. The superposition with the SVD algorithm (after best chain-to-chain mapping) results in a nearly perpendicular orientation of the two structures with respect to each other (Figure 4A,B), with an RMSD of 9.3 Å. Conversely, minimization of Φ results in an alignment of the first and last chain of the ladder, which run in the same direction, which seems to be a more sensible superposition. The RMSD corresponding to this superposition is 10.1 Å, i.e., not much higher than that obtained with SVD. Thus, as expected, LMAGDA finds the superposition with the largest matching fragment, given positioning the geometric centers of both structures in the same point.

Two other examples of the capability of LMAGDA to focus on alignable sections of the two compared structures are presented in Figure 6, in which selected pairs of MassiveFold models of T2234 (a trimer) and T2235 (a hexamer) are shown after superposing with the reference algorithm, which results in the same superposition as LMADA and LMAGDA, respectively. For both targets, the two structures differ mainly in the N-terminal section, these differences making a significant contribution to the RMSD, which is 10.6 Å for both the T2234 and T1235 models, as calculated with the reference algorithm or LMADA. As can be seen from panels A and C of the Figure, the reference algorithm and LMADA try to align the N-terminal sections at the expense of the alignment of the loops. (There are three loops for T2234, one of which is shown in the bottom of panels A and B, and six loops for T2235, all of which are shown in panels C and D.) Conversely, LMAGDA focuses on the alignable parts, while ignoring the non-alignable N-terminal sections, this resulting in a better alignment of the loops (panels B and D of the Figure). The last feature is also reflected in the $RMSD_{\Phi }$ values, which are 4.5 and 5.6 Åfor T2234 and T2235, respectively, these values being of the same order as those for pairs of structures superposing with a low RMSD (cf. Figure 4B). The overall RMSDs increase only to 10.7 Å.

From Table 1, it follows that LMAGDA gives worse results in terms of the difference between the RMSD ($RMSD_{d}$ in this case) and the reference RMSD and in terms of a higher percentage of pairs of structures wrongly assessed as dissimilar based on the 6 Å RMSD similarity cut-off. This feature of LMAGDA results from the minimization of a target function that is not aimed at reducing the largest distances between the parts of the superposed structures, in contrast to SVD. However, as demonstrated with the examples shown in Figure 5 and Figure 6, focusing on alignable sections can be of advantage in terms of obtaining a more sensible superposition. From Table 2, it follows that LMAGDA is about 1.5-fold more, on average, expensive in terms of timing than LMADA, which results from the minimization of Φ.

## 3. Materials and Methods

### 3.1. The Superposition Algorithms

We consider two structures of an ensemble of assemblies of *N* like molecules, each molecule containing *n* points (atoms or coarse-grained centers) to superpose. The distance between the two assemblies can be expressed as the RMSD between the corresponding points of the first and the second assembly, as given by Equation (Equation 6). In this work, we superpose the C^α^ atoms; thus, the RMSD is C^α^RMSD.(6)$$RMSD\left(x,y;P\right)=\sqrt{\frac{1}{nN}\sum_{k=1}^{N}\sum_{l=1}^{n}{∥x_{lk}-y_{lP_{k}}∥}^{2}}$$
where $x_{lk}$ and $y_{lk}$ are the vectors of the Cartesian coordinates of point *l* of molecule *k* of the first and of the second assembly, respectively, $P_{k}$ maps molecule *k* of the first assembly to the appropriate molecule of the second one, and P is the complete set of permutations that map the molecules of the first to those of the second assembly. For convenience, the assembly with coordinates x will be referred to as the reference assembly and that with coordinates y as the superposed assembly.

The optimal superposition problem can be formulated as the minimization of the sum of the squares of the distances of the two assemblies, *F*, in the rotation and translation of the coordinates of the superposed assembly and in the permutations P of the molecules of the superposed assembly, as expressed by Equation (Equation 7).(7)$$\min_{P,t,R}F\left(P,t,R\right)=\min_{P,t,R}\sum_{k=1}^{N}\sum_{l=1}^{n}{∥x_{lk}-\left(R\left(y_{lP_{k}}-\bar{y}\right)+t\right)∥}^{2}$$
where R is the rotation matrix, *t* is the translation vector, and $\bar{y}$ is the vector of the average coordinates of the superposed assembly [Equation (Equation 8)].(8)$$\bar{y}=\frac{1}{nN}\sum_{i=1}^{N}\sum_{j=1}^{n}y_{ji}$$

It can easily be shown that the translation vector minimizing *F* is equal to that of the average coordinates $\bar{x}$ of the reference assembly. Thus, if the coordinates of both assemblies are defined relative to their geometric centers, the translation vector minimizing *F* is a zero vector.

Given P, the minimization of *F* can be accomplished by using the SVD algorithm [4,5]. By running SVD for all possible molecule-to-molecule mappings (permutations), the global minimum of *F* can be found. The corresponding RMSD is the lowest RMSD. We term such a procedure the *reference algorithm*. Because it gives the lowest possible and, thus, clearly defined reference RMSD, we use it to assess the quality of the algorithms developed in this work. On the other hand, since the number of permutation rapidly grows with assembly size, this algorithm cannot be used for larger assemblies. Specifically, in our study, in which we had to gather meaningful statistics, oligomer size was effectively limited to 8 (see the Section 2).

To avoid dealing with the combinatorial problem mentioned above, we propose two algorithms, hereafter referred to as LMADA and LMAGDA, respectively. LMADA is based on finding a reasonable mapping (permutation) of the molecules of the superposed assembly on those of the reference assembly and running SVD given this single mapping. LMAGDA is based on minimizing a permutation-independent target function in order to find the optimal rotation matrix R. In what follows, we assume that the coordinates of the reference and those of the superposed assemblies are shifted to the respective geometric centers. The first two steps are common to LMADA and LMAGDA and will be described under LMADA. A block diagram summarizing the workflow of LMADA and LMAGDA is shown in Figure 7.

#### 3.1.1. LMADA



**Step 1.**



We express the rotations in terms of quaternion components $q=q_{0}+iq_{x}+jq_{y}+kq_{z}$, where $q_{0}$ is the real and $q_{x}$, $q_{y}$, and $q_{z}$ constitute the imaginary part of the quaternion. We construct a grid in quaternion components, varying $q_{0}$ from 0 to 1 and all other components from −1 to 1, with a step of 0.5. We exclude the grid point with all zero components (the zero-length quaternion). This gives $3\times 5^{3}-1=374$ grid points. At each point, we compute the normalized quaternion $\hat{q}=q/\left|q\right|$, where $\left|q\right|=\sqrt{q_{0}^{2}+q_{x}^{2}+q_{y}^{2}+q_{z}^{2}}$. The imaginary components of the normalized quaternion, ${\hat{q}}_{x}$, ${\hat{q}}_{y}$, and ${\hat{q}}_{z}$, define the direction of the Euler axis for the rotation, while ${\hat{q}}_{0}=cos\theta /2$, where θ is the angle for the rotation about the Euler axis; obviously, $\sin\theta /2=\sqrt{{\hat{q}}_{x}^{2}+{\hat{q}}_{y}^{2}+{\hat{q}}_{z}^{2}}$ if θ runs from $0^{\circ }$ to $360^{\circ }$. As can be seen from Figure 8, the quaternion coordinate grid defined in such a way covers the rotation space uniformly and fairly densely. Using the normalized quaternion coordinates, we then compute the rotation matrix R at each grid point, as given by Equation (Equation 9) [20].(9)$$R=\left[\begin{array}{ccc} {\hat{q}}_{0}^{2}+{\hat{q}}_{x}^{2}-{\hat{q}}_{y}^{2}-{\hat{q}}_{z}^{2} & 2({\hat{q}}_{x}{\hat{q}}_{y}-{\hat{q}}_{0}{\hat{q}}_{z}) & 2({\hat{q}}_{x}{\hat{q}}_{z}+{\hat{q}}_{0}{\hat{q}}_{y})\\ 2({\hat{q}}_{y}{\hat{q}}_{x}+{\hat{q}}_{0}{\hat{q}}_{z}) & {\hat{q}}_{0}^{2}-{\hat{q}}_{x}^{2}+{\hat{q}}_{y}^{2}-{\hat{q}}_{z}^{2} & 2({\hat{q}}_{y}{\hat{q}}_{z}-{\hat{q}}_{0}{\hat{q}}_{x})\\ 2({\hat{q}}_{z}{\hat{q}}_{x}-{\hat{q}}_{0}{\hat{q}}_{y}) & 2({\hat{q}}_{z}{\hat{q}}_{y}+{\hat{q}}_{0}{\hat{q}}_{x}) & {\hat{q}}_{0}^{2}-{\hat{q}}_{x}^{2}-{\hat{q}}_{y}^{2}+{\hat{q}}_{z}^{2} \end{array}\right]$$



**Step 2.**



We loop over all 374 points of the quaternion component grid. At a given grid point, we apply the rotation matrix defined by Equation (Equation 9) to calculate the rotated coordinates of the superposed assembly. Subsequently, we compute the distances between all molecules of the reference structure and those of the superposed structure. We define the distance between molecule *i* of the reference and molecule *j* of the superposed structure by Equation (Equation 10) (we recall that the coordinates of both assemblies are defined relative to their geometric centers, so no translation is needed).(10)$$d_{ij}=\sqrt{\frac{1}{n}\sum_{k=1}^{n}{∥x_{ki}-{\mathrm{Ry}}_{kj}∥}^{2}}$$

We sort the distances $d_{ij}$ from the smallest to the largest. Starting from the smallest distance, we exclude those for which one of the indices (*i* or *j*) has already appeared in the preceding elements of the list. This procedure gives *N* pairs of molecules, the first one from the reference assembly and the second one from the superposed assembly, with gradually increasing distances, starting from the smallest distance. By sorting the pairs according to the index of the molecule in the reference assembly, *i*, we obtain the complete mapping (permutation) $P_{1},P_{2},\ldots ,P_{N}$ of the molecules of the superposed assembly on those of the reference assembly. Subsequently, from the distances between the molecules of the reference assemblies and those of the superposed assembly mapped on them, we compute the RMSD, denoted as $RMSD_{d}$, as given by Equation (Equation 11).(11)$$RMSD_{d}=\sqrt{\frac{1}{N}\sum_{i=1}^{N}d_{iP_{i}}^{2}}$$

If the current $RMSD_{d}$ is smaller than the lowest $RMSD_{d}$ ($RMSD_{d}^{min}$) found for the previous grid points, we replace $RMSD_{d}^{min}$ with the current $RMSD_{d}$, and the corresponding permutation $P_{1}^{min},P_{2}^{min},\ldots ,P_{N}^{min}$ (for the next step of LMADA) and quaternion $q^{min}$ (for the next step of LMAGDA) with the current ones.

It should be noted that another algorithm for finding the optimal orientation, which requires a lower computational cost, has been designed by Helmich and Skierka [21]. Their algorithm is based on matching the largest dimensions of the two assemblies. It requires checking only 6 orientations (compared to 374 in ours) but does not yield sufficiently good results in superposing the assemblies of molecules. In the original work, it was applied to atom-to-atom and not molecule-to-molecule superposition, e.g., to superposing atomic clusters.



**Step 3.**



Given the mapping $P_{1}^{min},P_{2}^{min},\ldots ,P_{N}^{min}$, we use SVD to complete the superposition.

The cost of LMADA scales with $N^{2}$, as opposed to N!, for the brute-force permutation search.

#### 3.1.2. LMAGDA

LMADA is aimed at finding the lowest RMSD and is, therefore, good when we are interested in the overall fitting of the two structures. However, large distances will be taken care of in the first place, which is not necessarily desired when large assemblies are superposed. For large assemblies, the objective is to find the section(s) that superpose well on each other. Eliminating outlier molecules or their parts (as in the LGA [6,7] or the TM-score [8] algorithms) and retrying superposition with the reduced set of points is possible but could prove too expensive for larger assemblies. Therefore, we designed LMAGDA, which shares steps 1 and 2 with LMADA but, in step 3, employs a permutation-independent target function Φ(q) defined by Equation (Equation 12). This function, which is the negative of the logarithm of Gaussians in the distances of the atoms/sites of the molecules of the reference and the superposed structures, is minimized in quaternion components. The sum of the Gaussians in the differences in site–site distances of the compared structures and not their coordinates, termed *q*, was introduced earlier by Wolynes and coworkers [22]. Other functions with a maximum at the zero distance between the reference and the superposed points, e.g., the Lorentzian function that appears in the expression for TM-score [8], which is minimized in MM-align [11] and US-align [12], can also be used as an alternative to Gaussians. Φ(q) has multiple minima, and thus a good initial guess is required to start minimization. We use the quaternion corresponding to the lowest $RMSD_{d}$ found in step 2 (cf. LMADA).(12)$$\Phi \left(q\right)=-\ln\left[\sum_{i=1}^{N}\sum_{j=1}^{N}\sum_{k=1}^{n}\exp\left(-\frac{∥x_{ki}-R\left(q\right)y_{kj}{∥}^{2}}{2\sigma^{2}}\right)\right]$$
where σ is the standard deviation of the Gaussians. Φ can be interpreted as a measure of the overlap of the densities of the two assemblies, each density being represented as a sum of Gaussians centered on the points to superpose. Clearly, if the first and the second structures are identical save the permutation of molecules, Φ will reach the minimum for exact overlap with a small enough σ, regardless of permutation. For imperfect overlap, too small a σ can result in sticking the closest points of the superposed structure to those of the reference structure to maximize the respective Gaussians at the expense of the other overlaps. On the other hand, too a large σ makes the algorithm weakly sensitive to the differences between the two assemblies. In this work, we selected $\sigma =\sqrt{8}$, which is a value slightly lower than the C^α^⋯C^α^ virtual bond length for a trans peptide group equal to 3.8 Å. As opposed to RMSD, Φ of Equation (Equation 12) is more sensitive to well-matching sections of the two compared assemblies than to those that do not match.

To minimize Φ, we applied the Secant Unconstrained Minimization Solver (SUMSL) quasi-Newton algorithm [23]. Using an iterative minimization procedure, each step of which requires an $N^{2}$ effort, makes LMAGDA more expensive than LMADA, but its cost still scales with $N^{2}$.

It should be noted that the set of variables in minimizing Φ could be extended to add the components of the translation vector t of the superposed with respect to the reference assembly. However, in this work, we preferred to keep the centers of the two assemblies fixed in the same point in order to compare the results with those of applying LMADA and the reference algorithm. Moreover, this modification will presumably matter only when LMAGDA is extended to superposing ensembles of different numbers of like molecules.

Φ is usually negative and is dimensionless. Therefore, we transform it into $RMSD_{\Phi }$, defined by Equation (Equation 13), which has the dimension of the distance. The addition of $\ln(N^{2}n)$ assures that the argument of the square root function is not negative. For comparison with LMADA, we also calculate $RMDS_{d}$ [Equation (Equation 11)] corresponding to the optimized quaternion (and, thereby, the orientation of the superposed vs. the reference assembly).(13)$$RMSD_{\Phi }=\sqrt{2}\sigma \sqrt{\Phi +\ln(N^{2}n)}$$

### 3.2. Test Systems

We considered two kinds of test systems. The first one consisted of the 4-, 6-, 8-, and 10-chain assemblies of the 6-residue CysZ8 sequence (H-QAGIVV-NH_2_) The aggregation propensities of CysZ8 were studied in our earlier work [24] with the coarse-grained UNRES model of polypeptide chains [25]. In that work, we ran Multiplexed Replica Exchange Molecular Dynamics (MREMD) [26] simulations of the octamers of this peptide, using the UNRES implementation of MREMD [27]. We found that amyloid-like assemblies were formed.

For all four assemblies of CysZ8 mentioned above, we ran MREMD simulations with the NEWCT-9P variant of the UNRES force field [28] in a periodic box with a side corresponding to a concentration of about $2\times 10^{-3}$ M for each assembly. The other settings of the simulations were as in our earlier work [24]. Briefly, replicas were run at 12 temperatures ranging from 260 K to 370 K, each quadruplexed, which gives a total of 48 replicas. Each trajectory had 20,000,000 MD steps, with a 9.96 fs time step. After a run was completed, the Weighted Histogram Analysis Method (WHAM) [29] adapted to UNRES [30] was applied to determine the probabilities of the conformations at a desired temperature. We selected T=270 K to obtain a sufficient number of packed fibril-like structures. Subsequently, we selected the 100 structures with the highest probabilities to test the superposition algorithms. All pairs of structures (a total of 4950) were considered for up to 8-chain assemblies. For the 10-chain assemblies, the number of all chain permutations is 3,628,800. Therefore, the calculations would take too long to obtain the lowest possible (reference) RMSDs for all 4950 pairs, and only 10 structures (45 pairs) were considered. Because of poor statistics, these calculations were only used to compare the timings of different superposition algorithms.

The second test set consisted of the models of seven homooligomeric CASP16 targets produced and made available to the CASP16 predictors by the MassiveFold team [1]. MassiveFold uses AlphaFold [31] and ColabFold [32] to generate a large number of models. These were T2234 (trimer, 413 residues/chain), T2235 (hexamer, 115 residues/chain), T2238 (dimer, 327 residues/chain), T2240 (trimer, 653 residues/chain), T2249 (trimer, 488 residues/chain), T2259 (trimer, 243 residues/chain), and T2270 (hexamer, 437 residues/chain). For each target, the MassiveFold set contained models with different chain permutations. After the CASP16 experiment had been concluded, the experimental structures of T2234 and T1235 were released in the Protein Data Bank (PDB) [33]. These structures are parts of the 8qpq structure. No structure was obtained of T2238. The structures of the other 5 targets have not been released as yet, but their experimental structures are visualized in the Results section of the CASP16 web page at https://predictioncenter.org/casp16/results.cgi?tr_type=multimer, accessed on 1 March 2025. It should be noted that the MassiveFold models are not necessarily similar to these structures. For each of the proteins, we took every 100th MassiveFold model from the afm_basic set (a total of 100 structures, this giving 4950 pairs of structures to be superposed).

As mentioned at the beginning of Section 3.1, the selection of assemblies composed of small numbers of monomers was caused by the necessity of validating the new superposition algorithms against the reference algorithm, which involves carrying out SIVADE for all N! assignments of the molecules of the superposed to those of the reference assembly.

## 4. Discussion and Conclusions

We developed two algorithms, namely LMADA and LMAGDA, for the superposition of the assemblies of like molecules (e.g., peptide or protein homooligomers and homoaggregates), which scale as the square of the number of molecules in each assembly, as opposed to the factorial of this number for examining all possible mappings of the molecules of one assembly on those of the other one. Although the examples presented in this paper pertain to peptides and proteins, LMADA and LMAGDA are applicable to any other kind of molecule, e.g., nucleic acids and lipids. Both algorithms start from searching the rotations of one assembly relative to the other one on a grid of quaternion coordinates, which enables us to cover the rotation space uniformly. The comparison of the distances of the molecules of the first to those of the second assembly enables us to select the best mapping for each grid point and to find the best initial orientation of the molecules. In LMADA, the mapping giving the lowest estimated assembly-to-assembly distance [$RMSD_{d}$ of Equation (Equation 11)] is used to run the SVD. The quaternion corresponding to the lowest estimated distance is used as the starting point for the minimization of Φ [Equation (Equation 12)] in LMAGDA. The two algorithms were tested with the ensembles of the homooligomers of the CysZ8 hexapeptide studied in our earlier work [24], generated by means of UNRES/MREMD [27] simulations and with the models of the seven CASP16 [2] homooligomeric targets made available by the MassiveFold team [1].

LMADA was found to produce nearly as low an RMSD on average as the reference algorithm based on the enumeration of all possible permutations of chain indices leaving, at worst, only a few percent of pairs of structures with RMSD wrongly assessed to exceed the 6 Å RMSD similarity cut-off [17] (Table 1). In terms of timing, LMADA can be of advantage with respect to the reference algorithm starting from hexamers. Starting from octamers, the reference algorithm becomes impractical or even impossible to apply (Table 2). Therefore, LMADA appears to be suitable especially for cluster analysis (e.g., clustering the large body of all MassiveFold models of oligomers containing many like-chains), in which the calculation of the distances between the structures is a critical step.

LMAGDA does not give as good results as LMADA in terms of RMSD (Table 1), but this feature follows from the properties of the target function it minimizes [Equation (Equation 12)]. As opposed to RMSD, this function is the lowest for the superpositions resulting in a close alignment of parts of both assemblies, and those which are far apart contribute less to the sum of Gaussians. As shown in Figure 5 and Figure 6, such a superposition can give a more informative picture of the compatibility of the two structures than the distance-based superposition and better align the similar sections, without having to iteratively eliminate non-matching sections.

Both LMADA and LMAGDA can easily be generalized to assemblies consisting of several kinds of like molecules (e.g., A_x_B_y_ peptide or protein heterooligomers or aggregates of two kinds of peptide molecules) by mapping molecules of a given kind of the the first on those of the second assembly. This treatment can be extended to assemblies of molecules of different kinds, e.g., proteins and nucleic acids. LMAGDA, which finds the largest parts of the two assemblies superposing to each other well and is less sensitive to the non-superposing part, can also be generalized to superposing assemblies with a different number of like molecules in each of them. This extension would involve minimizing Φ of Equation (Equation 12) both in the quaternion and in the translation vector coordinates. It can also be used to align two protein structures to find the maximum overlap, which is usually carried out, e.g., by the LGA algorithm [6,7] in an iterative way by gradually eliminating the sections that do not overlap within a given cut-off. This possible future application can also be extended to align DNA/RNA molecules to find the well-overlapping segments.

## Figures and Tables

**Figure 1 molecules-30-01156-f001:**
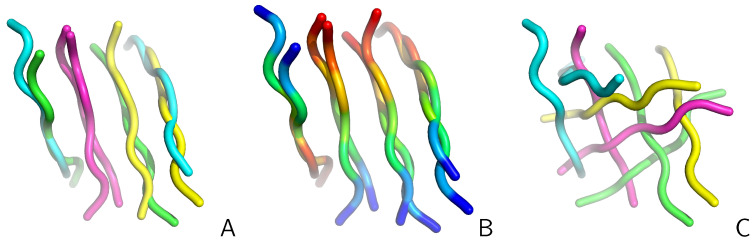
(**A**) Two structures of the CysZ8 tetramer simulated with UNRES superposed with the reference algorithm. The C^α^RMSD is 3.38 Å. The chains are colored as follows: A—green; B—blue; C—pink; and D—yellow. (**B**) as in (**A**) but each chain is colored from blue to red from the N- to the C-terminus. (**C**) Superposition of the two structures with the simple algorithm. The C^α^RMSD is 10.34 Å. The drawings were made with PyMOL [18].

**Figure 2 molecules-30-01156-f002:**
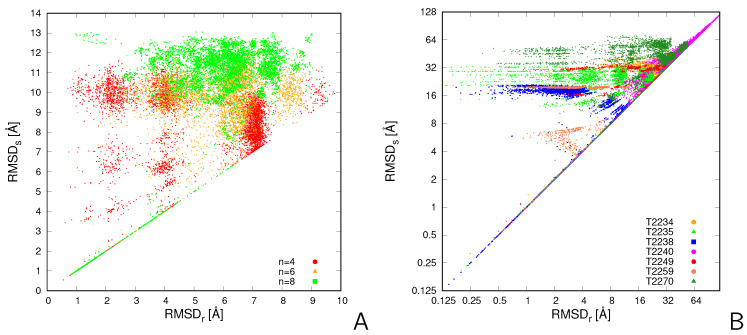
Plots of $RMSD_{s}$ vs. $RMSD_{r}$ for the 4950 pairs of (**A**) CysZ8 tetramer, hexamer, and octamer structures obtained in coarse-grained MREMD simulations with UNRES and (**B**) the MassiveFold models of 7 CASP16 homooligomeric targets. Due to a large RMSD span, a logarithmic scale with base 2 is used in panel (**B**). The plots were made with gnuplot [19].

**Figure 3 molecules-30-01156-f003:**
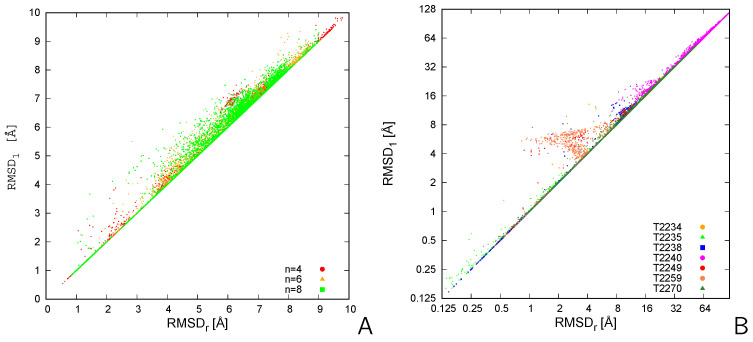
Plots of the $RMSD_{1}$ vs. $RMSD_{r}$ for the 4950 pairs of (**A**) the CysZ8 tetramers, hexamers, and octamers obtained in coarse-grained MREMD simulations with UNRES and (**B**) of MassiveFold models of the 7 CASP16 homooligomeric targets. Due to a large RMSD span, a logarithmic scale with base 2 is used in panel (**B**). The plots were made with gnuplot [19].

**Figure 4 molecules-30-01156-f004:**
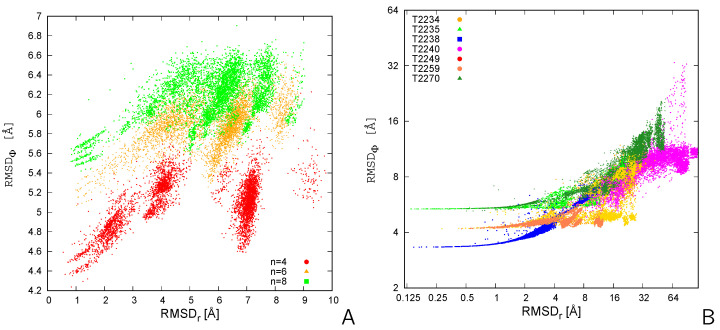
Plot of the $RMSD_{\Phi }$ vs. $RMSD_{r}$ for the 4950 pairs of (**A**) the CysZ8 tetramers, hexamers, and octamers obtained in coarse-grained MREMD simulations with UNRES and (**B**) the MassiveFold models of the 7 CASP16 homooligomeric targets. Due to a large RMSD span, a logarithmic scale with base 2 is used in panel (**B**). The plots were made with gnuplot [19].

**Figure 5 molecules-30-01156-f005:**
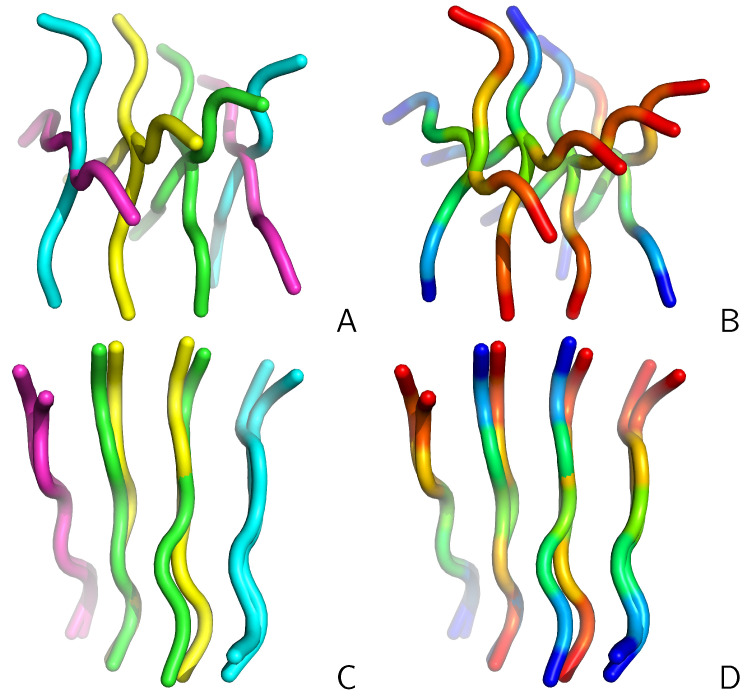
Two structures of the CysZ8 tetramer simulated with UNRES superposed by using LMAGDA, which in this case results in exactly the same superposition as the reference algorithm (**A**,**B**), and by using LMAGDA (**C**,**D**). The chains are colored differently in panels (**A**,**C**), and each chain is colored from blue to red from the N- to the C-terminus in panels (**B**,**D**). The chains are aligned similarly in both structures, but the directions of two of them are opposite. The reference algorithm and LMADA (which are strictly based on minimizing the distance) result in a perpendicular orientation of the two structures. Conversely, LMAGDA, which is based on the best overlap of the matching sections of the structures, results in a more sensible alignment. The drawings were made with PyMOL [18].

**Figure 6 molecules-30-01156-f006:**
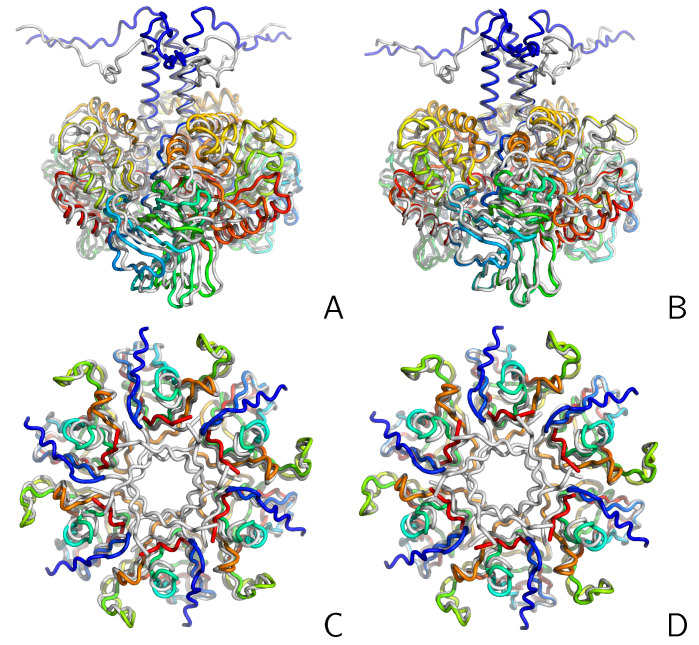
Selected pairs of the structures of MassiveFold models of T2234 (**A**,**B**) and T2235 (**C**,**D**) superposed with the reference algorithm (which gives the same results as LMADA in these cases) (**A**,**C**) and LMAGDA (**B**,**D**). The chains of the first structure in a pair are colored from blue to red from the N- to the C-terminus, while those of the other structure in a pair are in the light gray color. The views (side for T2234 and top for T2235, respectively) have been selected to emphasize the differences. The drawings were made with PyMOL [18].

**Figure 7 molecules-30-01156-f007:**
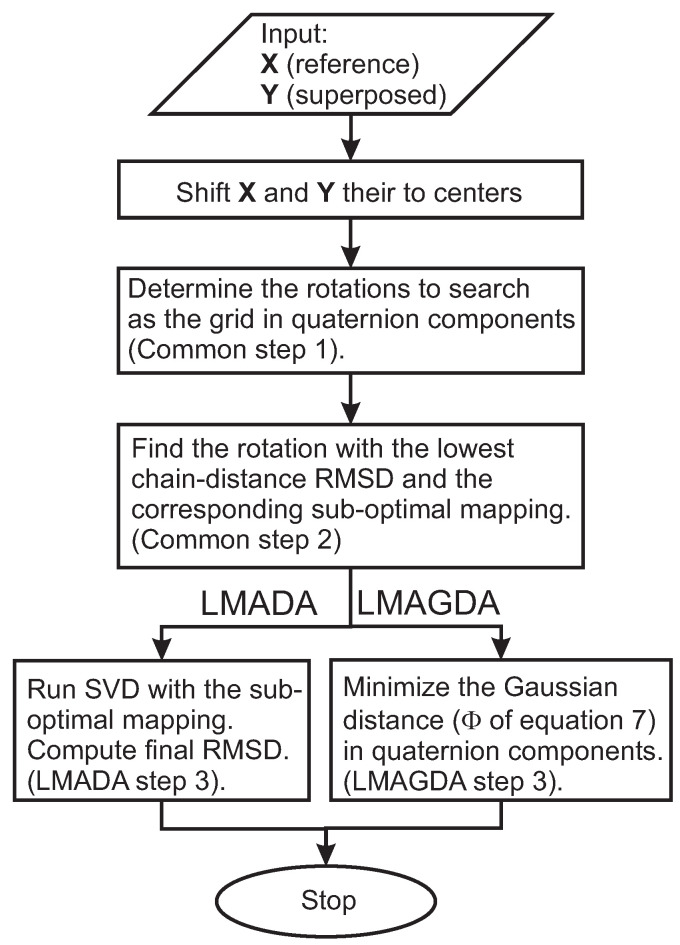
A block diagram summarizing the workflow of LMADA and LMAGDA.

**Figure 8 molecules-30-01156-f008:**
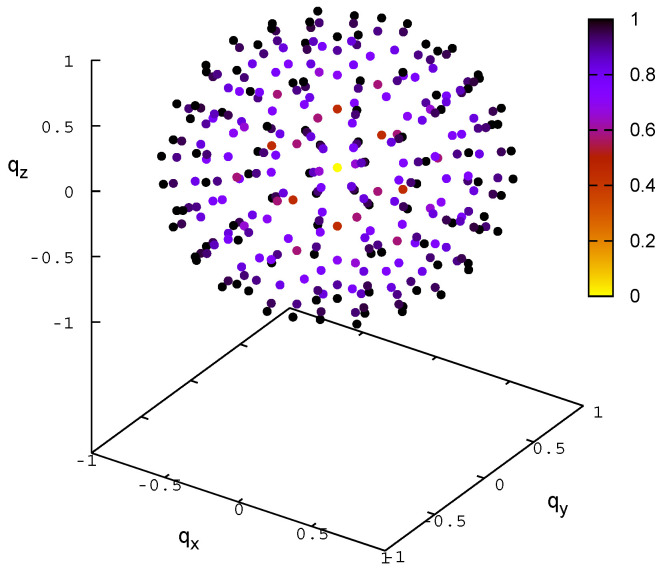
Visualization of the rotations obtained from the grid on quaternion coordinates. The tops of the Euler axes scaled by sinθ/2, where θ is the rotation angle about the axis, are shown as small colored spheres. Points farther from the center correspond to larger rotation angles about the Euler axes. Additionally, the spheres are colored from yellow to dark purple, depending on the value of sinθ/2. The plot was made with gnuplot [19].

**Table 1 molecules-30-01156-t001:** Comparison of the results of superposing the pairs of structures of the systems studied in this work by using the simple algorithm, LMADA, and LMAGDA with those of using the reference algorithm.

System	${\bar{\Delta \mathrm{RMSD}}}_{s}$ ^1^	$\bar{\left(\frac{\Delta {\mathrm{RMSD}}_{s}}{{\mathrm{RMSD}}_{r}}\right)}$ ^1^	$\%R_{s}$ ^1^
# Chains (*N*)	${\bar{\Delta \mathrm{RMSD}}}_{1}$	$\bar{\left(\frac{\Delta {\mathrm{RMSD}}_{s}}{{\mathrm{RMSD}}_{r}}\right)}$	$\%R_{1}$
# res./Chain (*n*)	${\bar{\Delta \mathrm{RMSD}}}_{d}$	$\bar{\left(\frac{\Delta {\mathrm{RMSD}}_{d}}{{\mathrm{RMSD}}_{r}}\right)}$	$\%R_{d}$
CysZ8 × 4	3.29 (2.75)	1.10 (1.52)	84.3
4	0.04 (0.12)	0.01 (0.04)	0.5
6	0.13 (0.19)	0.04 (0.06)	0.7
CysZ8 × 6	3.83 (1.74)	0.77 (0.62)	93.8
6	0.10 (0.19)	0.02 (0.04)	4.1
6	0.59 (0.67)	0.11 (0.17)	27.7
CysZ8 × 8	4.92 (1.93)	0.92 (0.72)	87.1
8	0.12 (0.23)	0.02 (0.06)	6.5
6	0.60 (0.76)	0.11 (0.15)	28.8
T2234	4.35 (7.18)	0.29 (0.80)	11.3
3	0.07 (0.26)	0.00 (0.05)	3.8
413	2.68 (3.54)	0.12 (0.30)	7.5
T2235	13.83 (8.38)	5.47 (15.3)	94.8
6	0.01 (0.04)	0.00 (0.04)	0.0
115	0.70 (1.37)	0.04 (0.08)	0.0
T2238	6.02 (7.63)	3.16 (6.02)	46.2
2	0.05 (0.31)	0.01 (0.07)	0.1
327	0.32 (0.67)	0.06 (0.16)	0.4
T2240	2.81 (3.25)	0.09 (0.19)	11.1
3	0.40 (0.97)	0.01 (0.05)	0.0
653	14.15 (22.6)	0.29 (0.50)	11.1
T2249	4.92 (1.93)	0.92 (0.72)	87.1
3	0.04 (0.29)	0.01 (0.15)	1.9
488	2.63 (5.22)	0.12 (0.25)	3.0
T2259	4.92 (1.93)	0.92 (0.72)	87.1
3	0.20 (0.75)	0.08 (0.38)	3.4
243	0.29 (0.79)	0.10 (0.38)	4.4
T2270	4.92 (1.93)	0.92 (0.72)	87.1
6	0.07 (0.27)	0.01 (0.10)	2.0
437	5.69 (7.48)	0.22 (0.34)	11.4

^1^ The top, middle, and bottom values correspond to the simple superposition (chains with the same IDs of the superposed and the reference structures matched), LMADA, and LMAGDA, respectively. The standard deviations of the RMSD differences from those of the reference algorithm are in parentheses.

**Table 2 molecules-30-01156-t002:** Comparison of CPU times for the reference algorithm, LMADA, and LMAGDA.

System	# Chains (*N*)	# res./Chain (*n*)			
			CPU Time/Pair [ms] ^1^		
			Ref.	LMADA	LMAGDA
CysZ8 × 4	4	6	0.0457 [0.0441, 0.0467]	0.344 [0.337, 0.351]	0.396 [0.384, 0.404]
CysZ8 × 6	6	6	0.936 [0.913, 0.952]	0.988 [0.968, 1.01]	1.10 [1.07, 1.12]
CysZ8 × 8	8	6	62.3 [60.5, 63.2]	2.07 [2.03, 2.12]	2.25 [2.19, 2.29]
CysZ8 × 10	10	6	6605 [6490, 6677]	3.71 [3.62, 3.75]	3.92 [3.81, 3.98]
T2234	3	413	0.172 [0.170, 0.174]	6.31 [6.17, 6.44]	8.92 [8.92, 8.93]
T2235	6	115	10.0 [9.89, 10.1]	6.32 [6.12, 6.42]	8.05 [8.05, 8.06]
T2238	2	327	0.0496 [0.0490, 0.0499]	2.65 [2.57, 2.69]	3.52 [3.52, 3.53]
T2240	3	653	0.260 [0.259, 0.260]	10.1 [9.96, 10.1]	15.3 [15.3, 15.3]
T2249	3	488	0.200 [0.199, 0.200]	7.58 [7.58, 7.59]	9.87 [9.85, 9.88]
T2259	3	243	0.111 [0.111, 0.111]	3.80 [3.80, 3.80]	4.79 [4.78, 4.80]
T2270	6	437	37.1 [37.0, 37.2]	22.4 [22.4, 22.4]	31.0 [31.0, 31.0]

^1^ CPU times with a single Intel X86 64-bit processor. The values without brackets are averages over 3 independent runs, the values in square brackets are minimum and maximum values. The source code was compiled with the ifort version 19.1.1.217 Intel Fortran compiler, with the regular (O_2_) optimization. The Fortran CPU_TIME function was used, and only the time for superposing the structures with the method of choice was measured.

## Data Availability

The FORTRAN code of the algorithm together with the examples presented in this paper are available at https://unres.pl, accessed on 1 March 2025. The original contributions presented in this study are included in the Appendix A.

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
