# Peer review of "Two Methods for Superposing the Structures of Like-Molecule Assemblies: Application to Peptide and Protein Oligomers and Aggregates"

_molecules, 2025, doi:10.3390/molecules30051156_

Round 1
Reviewer 1 Report
Comments and Suggestions for Authors
This manuscript presents two efficient algorithms (Algorithm 1 and Algorithm 2) designed for the superimposition of Like-Molecule assembly structures, addressing the computational bottleneck associated with traditional exhaustive alignment methods, which exhibit N! complexity. Algorithm 1 focuses on optimizing overall structural alignment, whereas Algorithm 2 emphasizes the alignment of geometrically matched regions by minimizing the Gaussian overlap function. This approach sacrifices some global RMSD accuracy to enhance local structural consistency. Experimental results demonstrate that Algorithm 1 achieves an RMSD close to that of the exhaustive method (average difference ≤ 0.2 Å), with computational cost scaling as N², resulting in significant efficiency improvements for tetramer and larger systems. Algorithm 2, while exhibiting a slightly higher RMSD, demonstrates greater sensitivity to local matching, making it particularly suitable for scenarios where local structural similarity is of primary interest.
The study offers a valuable tool for the comparison of large-scale like-molecular assemblies, with notable potential for applications involving the comparison of simulated/predicted and experimental structures.
While I am not an expert in computational biology and thus refrain from evaluating the technical intricacies of the algorithms, I will focus on their practical implications. The reduction in computational complexity from N! to N² is indeed remarkable. However, several issues limit the article's accessibility and broader applicability, particularly for readers outside the field:
- Algorithm Naming and Differentiation: The authors propose two distinct algorithms tailored for different scenarios. However, the current naming convention ("Algorithm 1" and "Algorithm 2") does not adequately reflect their unique characteristics. It would be beneficial to assign more descriptive names that highlight their respective functionalities and intended use cases.
- Validation Dataset: The validation primarily relies on the CysZ8 multimer and a limited selection of seven targets from CASP16, with the highest oligomerization state being a hexamer. While these targets provide some insight, they do not sufficiently represent the complexity of real-world Like-Molecule assemblies, such as filamentous proteins, amyloid proteins, or viral capsids. These macromolecules, which have seen significant advancements in resolution in recent years, often consist of hundreds of identical peptide chains and represent critical applications where efficient superposition algorithms are most needed. Expanding the validation dataset to include such complex structures would significantly enhance the article's applicability and impact.
- Visualization of Results: Figures 2 and 6 effectively illustrate the performance of the algorithms on CysZ8 multimers. However, for other real protein targets, the authors should consider including additional visual representations of the superposition results. Such visualizations would provide a more intuitive understanding of the differences between the algorithms and their respective strengths in handling various structural complexities.
In summary, while the manuscript presents a promising advancement in the field of structural alignment, addressing these points would improve its clarity, applicability, and overall value to a broader audience.
Comments on the Quality of English LanguageGood
Author Response
We would like to thank this Reviewer for his/her time and effort of evaluating our manuscript and for helpful feedback. We revised the manuscript accordingly. Our point-to-point answers to Reviewers’ criticism and the description of the revisions made are below. We also provide a markup version of the manuscript with changes in red font.
Comment 1:
Algorithm Naming and Differentiation: The authors propose two distinct algorithms tailored for different scenarios. However, the current naming convention ("Algorithm 1" and "Algorithm 2") does not adequately reflect their unique characteristics. It would be beneficial to assign more descriptive names that highlight their respective functionalities and intended use cases.
Reply:
We renamed Algorithm 1 and 2 to LMADA (Like-Molecule Assembly Distance Alignment algorithm) and LMAGDA (Like-Molecule Assembly Gaussian Distance Alignment algorithm), respectively. These names capture the essence of the algorithms. Changing algorithms’ names implied in revision of the text of the Abstract in lines 6-11 and of the Introduction in lines 61-62.
Comment 2:
Validation Dataset: The validation primarily relies on the CysZ8 multimer and a limited selection of seven targets from CASP16, with the highest oligomerization state being a hexamer. While these targets provide some insight, they do not sufficiently represent the complexity of real-world Like-Molecule assemblies, such as filamentous proteins, amyloid proteins, or viral capsids. These macromolecules, which have seen significant advancements in resolution in recent years, often consist of hundreds of identical peptide chains and represent critical applications where efficient superposition algorithms are most needed. Expanding the validation dataset to include such complex structures would significantly enhance the article's applicability and impact.
Reply: The validation data set was selected to enable us comparing the new algorithms with the reference algorithm, which is base on exhaustive search of all N! permutations of molecules. This requirement precludes anything larger than octamer. Including larger assemblies would enable us only to run the new algorithms, without the possibility of comparing the calculated RMSD with the reference (lowest) RMSD. We have added the respective comment in the revised manuscript (page 14, lines 385-388).
Comment 3:
Visualization of Results: Figures 2 and 6 effectively illustrate the performance of the algorithms on CysZ8 multimers. However, for other real protein targets, the authors should consider including additional visual representations of the superposition results. Such visualizations would provide a more intuitive understanding of the differences between the algorithms and their respective strengths in handling various structural complexities.
Reply: We added the new Figure 6, which compares selected pairs of the MassiveFold models of T2234 and T2235, aligned with the reference algorithm/LMADA (formerly Algorithm 1) and with LMAGDA (formerly Algorithm 2) and added the pertinent discussion (page 8, lines 182-197 and page 15, lines 421-424). The structures in both pairs differ by the N-terminal sections and using LMAGDA results in a better alignment of the other alignable sections, especially the loops.
Comment 4:
In summary, while the manuscript presents a promising advancement in the field of structural alignment, addressing these points would improve its clarity, applicability, and overall value to a broader audience.
Reply: We thank the Reviewer for helpful comments and hope that we have addressed the above points satisfactorily.
Reviewer 2 Report
Comments and Suggestions for Authors
The authors present a very interesting study on efficient algorithms for the superposition of molecular assemblies, addressing the computational challenges associated with exhaustive permutation searches. Two algorithms have been developed, leveraging quaternion-based orientation search and optimization techniques to achieve accurate structural alignment with significantly reduced computational cost. The proposed methods offer valuable applications in protein structure analysis, clustering, and model comparison. However, several important concerns must be addressed before the manuscript is suitable for publication.
- The key differences between MassiveFold and AlphaFold must be clarified, and the absence of a direct comparison with the latest AlphaFold version should be addressed.
- Clarify when Algorithm 2 should be used despite its higher RMSD.
- Although Algorithm 1 is stated to be advantageous from octamers onwards, a more detailed runtime comparison is needed to better assess its feasibility.
- The scaling of the new algorithms is impressive, but validation with actual execution times and hardware specifications is needed.
- The coordinates of all systems and raw PyMOL files must be provided in the Supporting Information to ensure reproducibility and facilitate further analysis.
I recommend the paper for publication, provided the concerns are properly addressed to enhance its robustness and completeness. Addressing these issues will strengthen the manuscript and its scientific contribution.
Author Response
We would like to thank this Reviewer for his/her time and effort of evaluating our manuscript and for helpful feedback. We revised the manuscript accordingly. Our point-to-point answers to Reviewers’ criticism and the description of the revisions made are below. We also provide a markup version of the manuscript with changes in red font.
Comment 1:
- The key differences between MassiveFold and AlphaFold must be clarified, and the absence of a direct comparison with the latest AlphaFold version should be addressed.
Reply:
MassiveFold uses AlphaFold and ColabFold to generate decoys on a massive basis. We have included this explanation in the revised manuscript (page 14 lines 371-372).
Comment 2:
- Clarify when Algorithm 2 should be used despite its higher RMSD.
Reply:
In lines 367-373 on page 13 of the original manuscript we pointed out that application of Algorithm 2 (now LMAGDA) results in a closer alignment of sections of the superposed assemblies, e.g., the loops. Therefore, if one wants to focus on aligning the largest sections of the two assemblies that can be aligned closely, Algorithm 2 (LMAGDA) should be used instead of Algorithm 1 (now LMADA) which, as the reference algorithm, minimizes the average distance between the two assemblies. It should be noted that the main contribution to a higher RMSD obtained by LMAGDA compared to LMADA and the reference algorithm results from the contributions of the segments that cannot be aligned well and which are largely ignored by LMAGDA. We have expanded the Discussion and Conclusions section to add this remark (page 15, lines 421-424).
Comment 3:
- Although Algorithm 1 is stated to be advantageous from octamers onwards, a more detailed runtime comparison is needed to better assess its feasibility.
Reply:
We have provided more details in the revised Table 2 (maximum, minimum, and average execution times). All times were averages over 3 independent runs. Please note than we have optimized the algorithms since the first submission and the execution times are slightly smaller now. We revised the text following the new timing data (page 6, lines 156-160 and page 9, lines 203-206).
Comment 4:
- The scaling of the new algorithms is impressive, but validation with actual execution times and hardware specifications is needed.
Reply:
The timing was measured with the Fortran CPU_TIME function and only the computation part was timed. This information was included in the revised footnote to Table 2.
Comment 5:
- The coordinates of all systems and raw PyMOL files must be provided in the Supporting Information to ensure reproducibility and facilitate further analysis.
Reply:
We included the coordinates of all systems studied in the PDB format and the PyMol dumps corresponding to Figures 1, 5 and the new Figure 6 in the Supporting Information. Please note that the figure numbering changed following our complaining with Editor’s request to move the Materials and Methods section after Results. The software and test data (the CysZ8 oligomers) are available at https://unres.pl in the Downloads section in the gaussfit.2.0.tar.gz archive.
Comment 6:
I recommend the paper for publication, provided the concerns are properly addressed to enhance its robustness and completeness. Addressing these issues will strengthen the manuscript and its scientific contribution.
Reply:
We thank the Reviewer for helpful comments, which enabled us to improve the manuscript and hope that we have addressed them satisfactorily.
Reviewer 3 Report
Comments and Suggestions for Authors
Liwo and Lesniewski introduce two new algorithms for superimposition of molecular assemblies. Their main advantage is that there is no need to analyze all permutations of the chain superimpositions to find a satisfactory solution. The first one applies SVD (singular value decomposition) approach, while the other minimizes the distance function in quaternion components. The algorithms are evaluated on a set of peptides and proteins. The authors highlight the polynomial computational complexity of both approaches and high quality of the results as compared to exhaustive method.
The paper is written relatively well, although many sentences are too long and hard to follow. Therefore, I recommend that the authors re-check the text and split too long sequences. Other comments are provided below.
1) CASP is a well-known initiative for evaluating predicted molecular structures, though it is primarily protein-centric. I kindly suggest that the authors familiarize themselves with other structure superimposition and alignment algorithms designed for non-protein assemblies. While I do not require computational experiments using these methods, mentioning them in the Introduction would provide a more comprehensive overview of state-of-the-art approaches. Examples of such algorithms can be found in the following papers: Dror et al. (doi: 10.1093/nar/gkl312), Hoksza et a. (doi: https://doi.org/10.1093/bioinformatics/bts3019), Zurkowski et al. (doi: 10.1093/bioinformatics/btad315; 10.1093/nar/gkae259). Additionally, I am interested in the possibility of using Algorithm 1 and Algorithm 2 for non-protein complexes or assemblies that contain proteins and other molecules. Is it possible to do it?
2) "The distance between the two assemblies can be expressed as the RMSD between points of the molecules of the first assembly and the matching molecules of the second assembly, respectively, as given" -> "The distance between the two assemblies can be expressed as the RMSD between the corresponding points of the first and the second assembly, as given"
3) In formula (1), RMSD should be replaced with RMSD(k,Pk)
4) "This is not feasible for larger assemblies" - what is meant by a larger assembly here? Is it about the numer of atoms in all melecules of the assembly or the numer of molecules in the assembly? And which size (in the numer of atoms / molecules) is still feasible and which is not?
5) Adding a figure with workflows of both algorithms would improve the readability. I strongy suggest to prepare such schemes.
6) I think, I must have missed the refereces to the "reference algorithm" used in benchmarking. Is it one of the algorithms used in CASP16? If not, I recommend trying Algorithm 1 and Algorithm 2 against one of the CASP methods. For example, when a new prediction method is introduced, it is a gold standard to benchmark it on CASP targets and compare to other methods used in CASP. I suggest it should be a similar standard with the scoring methods.
7) How would the ranking of groups/models in CASP16 change if the organizers used one of the algorithms presented in the paper?
Comments on the Quality of English Language
Re-check the paper, many sentences are too long and hard to follow.
Author Response
We would like to thank this Reviewer for his/her time and effort of evaluating our manuscript and for helpful feedback. We revised the manuscript accordingly. Our point-to-point answers to Reviewers’ criticism and the description of the revisions made are below. We also provide a markup version of the manuscript with changes in red font.
Comment 1:
Liwo and Lesniewski introduce two new algorithms for superimposition of molecular assemblies. Their main advantage is that there is no need to analyze all permutations of the chain superimpositions to find a satisfactory solution. The first one applies SVD (singular value decomposition) approach, while the other minimizes the distance function in quaternion components. The algorithms are evaluated on a set of peptides and proteins. The authors highlight the polynomial computational complexity of both approaches and high quality of the results as compared to exhaustive method.
The paper is written relatively well, although many sentences are too long and hard to follow. Therefore, I recommend that the authors re-check the text and split too long sequences. Other comments are provided below.
Reply:
We split too long sentences into shorter sentences (page 2, lines 35-38 and 71-73; page 4, lines 122-124; page 6, lines 152-156; pages 8 and 9, lines 173-177; lines 5-8 of the legend of Figure 5 in page 8; page 9, lines 202-206; page 13, lines 304-308; page 14, lines 349-352, 358-363 and 363-366).
Comment 2:
1) CASP is a well-known initiative for evaluating predicted molecular structures, though it is primarily protein-centric. I kindly suggest that the authors familiarize themselves with other structure superimposition and alignment algorithms designed for non-protein assemblies. While I do not require computational experiments using these methods, mentioning them in the Introduction would provide a more comprehensive overview of state-of-the-art approaches. Examples of such algorithms can be found in the following papers: Dror et al. (doi: 10.1093/nar/gkl312), Hoksza et a. (doi: https://doi.org/10.1093/bioinformatics/bts3019), Zurkowski et al. (doi: 10.1093/bioinformatics/btad315; 10.1093/nar/gkae259). Additionally, I am interested in the possibility of using Algorithm 1 and Algorithm 2 for non-protein complexes or assemblies that contain proteins and other molecules. Is it possible to do it?
Reply:
We thank the Reviewer for pointing to the references to large RNA structure alignment algorithms. The problem of finding closely matching sections of two large RNA molecules is related to like-molecule-assembly alignment. We have included this statement and cited the suggested references (new references 13-15), as well references to protein-alignment algorithm (new references 11 and 12) along with a brief discussion in the Introduction section (page 2 lines 49-59).
Both algorithms developed in this work can be used to align any assemblies of like-molecules, as can be gathered from the title of the paper, which points to like-molecule assemblies in general. Moreover, in the first sentence of the Discussion and Conclusions section of the original manuscript (page 12, lines 344-347; page 15, lines 390-394 of the revised manuscript) it was stated that peptides and proteins were examples of possible applications. In the revised manuscript, we stated the general applicability of our algorithms explicitly (page 15, lines 394-396). As we remarked in the Discussion and Conclusions section (page 13, lines 374-377 of the original manuscript; page 15, lines 425-428 of the revised manuscript), the algorithm can easily be extended assemblies containing several kinds of like molecules, which covers protein-nucleic acid complexes. We added the latter remark to the revised manuscript (page 15, lines 428-429). It seems that modified Algorithm 2 (now LMAGDA) could be applied DNA/RNA superposition problem, as well as, in general, to aligning two systems to find the largest closely-similar sections. We have already mentioned its application to alignment as an alternative to LGA (page 13, lines 301-304 of the original manuscript; page 15, lines 433-436 of the revised manuscript). We have now expanded this sentence to include large DNA/RNA molecules (page 15, lines 436-437).
Comment 3:
2) "The distance between the two assemblies can be expressed as the RMSD between points of the molecules of the first assembly and the matching molecules of the second assembly, respectively, as given" -> "The distance between the two assemblies can be expressed as the RMSD between the corresponding points of the first and the second assembly, as given"
Reply:
Corrected (page 9, lines 212-213).
Comment 4:
3) In formula (1), RMSD should be replaced with RMSD(k,Pk)
Reply:
k is the index of summation. We changed the left-hand side of Equation (1) [now Equation (6)] to RMSD(x,y;P) to indicate the quantities the RMSD depends on. We also revised the text after Equation (6) (page 9, lines 216-218).
Comment 5:
4) "This is not feasible for larger assemblies" - what is meant by a larger assembly here? Is it about the numer of atoms in all melecules of the assembly or the numer of molecules in the assembly? And which size (in the numer of atoms / molecules) is still feasible and which is not?
Reply:
“Larger” means the number of molecules in an assembly greater than 8, as stated in page 10, lines 232-238 of the revised manuscript and reiterated the justification of the selection of test systems in the new paragraph in page 14, lines 388-388. As follows from the timings collected in Table 2, for N=10 alignment with exhaustive enumeration takes over 6 seconds per pair, which is not acceptable.
Comment 6:
5) Adding a figure with workflows of both algorithms would improve the readability. I strongy suggest to prepare such schemes.
Reply:
Done as requested (New Figure 7). Please note that the numbers of all Figures changed following our complying with Editor’s request to place the Materials and Methods section after Results.
Comment 7:
6) I think, I must have missed the refereces to the "reference algorithm" used in benchmarking. Is it one of the algorithms used in CASP16? If not, I recommend trying Algorithm 1 and Algorithm 2 against one of the CASP methods. For example, when a new prediction method is introduced, it is a gold standard to benchmark it on CASP targets and compare to other methods used in CASP. I suggest it should be a similar standard with the scoring methods.
Reply:
The “reference algorithm” is exhaustive enumeration of all possible molecule-to-molecule mappings of the reference and superposed structures, running SVD to compute RMSD for each mapping, and selecting the lowest RMSD. It has been described explicitly in page 7, lines 228-232 of the original manuscript. In the revised manuscript, we included a formal definition of the reference algorithm along with explanation why it should be used for comparison in Methods after Equation 2, which is Equation 7 in the revised manuscript (page 10, lines 232-239) and have modified the beginning of the Results section (page 2, lines 75-80) to make this term clear. While the references cited in our paper mention exhaustive enumeration of all molecule-to-molecule mapping, this procedure is not defined there as a “reference algorithm”. Therefore, we did not cite any reference to our “reference algorithm”.
In CASP, monomer interfaces in a model are compared against those in the reference structure, the key measures are Interface Patch Similarity (IPS) and Interface Contact Similarity (ICS) [Lafita et al., Proteins 86 (2018) 247–256, which is Reference 16 in the revised manuscript]. We included this remark in the Introduction section of the revised manuscript (page 2, lines 55-59). Therefore, our algorithms cannot be compared with those that are used in CASP.
As we pointed out in Discussions and Conclusions (page 13, lines 381-384 of the original and page 14 lines 432-436 of the revised manuscript), Algorithm 2 (now LMAGDA) could be considered as an alternative to LGA (and compared with it) for the alignment of single protein molecules to find the largest overlapping fragment. However, this extension is beyond the scope of this paper, which is focused on the superposition of like-molecule assemblies, including protein homooligomers and homoaggregates.
Comment 8:
7) How would the ranking of groups/models in CASP16 change if the organizers used one of the algorithms presented in the paper?
Reply:
As pointed out in our reply to Reviewer comment 7, CASP uses other measures of oligomer similarity and the oligomers, which focus on monomer interface rather than on whole-structure compatibility (page 2, lines 55-59). Overall RMSD is only a secondary measure. Therefore, we do not expect that rankings would change much given the present evaluation rules. Our algorithm could certainly be used to compute the overall RMSD and LMAGDA could additionally be used